# Polymer Distribution and Mechanism Conversion in Multiple Media of Phase-Separated Controlled-Release Film-Coating

**DOI:** 10.3390/pharmaceutics11020080

**Published:** 2019-02-14

**Authors:** Lu Chen, Guobao Yang, Xiaoyang Chu, Chunhong Gao, Yuli Wang, Wei Gong, Zhiping Li, Yang Yang, Meiyan Yang, Chunsheng Gao

**Affiliations:** State key Laboratory of Toxicology and Medical Countermeasures, Beijing Institute of Pharmacology and Toxicology, Beijing 100850, China; c670406@163.com (L.C.); yangguobao0101@163.com (G.Y.); cxy15010248773@163.com (X.C.); 13521775490@163.com (C.G.); wangyuli764@126.com (Y.W.); usnitro2004@126.com (W.G.); dearwood2010@126.com (Z.L.); jiamusi101@126.com (Y.Y.)

**Keywords:** release media, hydroxypropyl cellulose, ethyl cellulose, microstructure, film-coating, release mechanism

## Abstract

Phase-separated films of water-insoluble ethyl cellulose (EC) and water-soluble hydroxypropyl cellulose (HPC) can be utilized to tailor drug release from coated pellets. In the present study, the effects of HPC levels and the pH, type, ionic strength and osmolarity of the media on the release profiles of soluble metoprolol succinates from the EC/HPC-coated pellets were investigated, and the differences in drug-release kinetics in multiple media were further elucidated through the HPC leaching and swelling kinetics of the pellets, morphology (SEM) and water uptake of the free films and the interaction between the coating polymers and the media compositions. Interestingly, the drug release rate from the pellets in different media was not in agreement with the drug solubility which have a positive correlation with the drug dissolution rate based on Noyes–Whitney equation law. In particular, the drug release rate in acetate buffer at pH 4.5 was faster than that in other media despite the solubility of drug was relatively lower, regardless of the HPC levels. It may be attributed to the mutual effect between the EC and acetate buffer, which improved the permeability of the film. In contrast, the release of drug in HCl solution was dependent on the HPC levels. Increasing the levels of HPC increased the effects of hydrogen ions on the polymer of HPC, which resulted in a lower viscosity and strength of the gel, forming the larger size of pores in polymer films, thus increasing the drug diffused from the coating film. Further findings in phosphate buffer showed a reduction in the drug release compared to that in other media, which was only sensitive to the osmolarity rather than the HPC level and pH of the buffer. Additionally, a mathematical theory was used to better explain and understand the experimentally measured different drug release patterns. In summary, the study revealed that the effects of the media overcompensated that of the drug solubility to some extent for controlled-release of the coating polymers, and the drug release mechanism in multiple media depend on EC and HPC rather than on HPC alone, which may have a potential to facilitate the optimization of ideally film-coated formulations.

## 1. Introduction

Polymer blend coatings of water-insoluble ethyl cellulose (EC) and water-soluble hydroxypropyl cellulose (HPC) can be utilized to control drug release from pellets in oral solid-dosage forms, in which HPC act as a pore-former [1,2,3,4,5,6]. When exposed to aqueous solution, the mass transport of the EC/HPC film-coated pellets is illustrated in Figure 1. The process always includes the following steps: (1) aqueous fluids penetrating into the system; (2) polymer swelling; (3) drug dissolving and pore-former leaching; (4) pores/channels forming in the film or the osmosis-driven water influx into the system, the hydrostatic pressure is built inside the pellets; and (5) gradual drug release into the bulk fluid over time by diffusion, osmotic processes, or a combination of the two mechanisms [7,8,9]. Therefore, from the above process, it is revealed that the drug release is influenced by many factors—such as manufacturing conditions [10] and formulation variables including drug solubility [11] molecular weight [1,2,12,13], and the level of coating polymer [3,5,6,14,15]—which have been extensively reported on in recent years. 

Additionally, the drug release patterns were also affected by the in vitro test conditions. In addition to the rotation speed and temperature, the pH of the medium [16,17], the buffer composition and capacity [18,19], and the ionic strength or osmolarity of the release media are also important to the drug release [20,21,22]. To simulate the physiological conditions of the GI tract, in vitro release media containing HCl, acetate, and phosphate in the pH range of 1~7.6 are commonly used [18]. Wesseling and Bodmeier reported that the higher degree of dissociation of the sodium lauryl sulfate (which is present in the aqueous-based EC dispersion) in pH 7.4 buffer resulted in the faster drug release compared to that in 0.1 M HCl [23]. In addition, the release of drug is increased sharply in pH 1.16 and pH 7.45 buffer solutions compared with that in water, owing to the depolarization of the polymer HPMC and, thus, lowering of the viscosity and strength of the film [24]. Interestingly, even at the same pH of 5.4, the drug release of quinidine gluconate was more faster in acetate buffer than in phosphate buffer [25]. Meanwhile, the ionic strength of the media also has an effect on the HPC system. The degree of hydration and swelling of HPC was decreased as the ionic strengths increased from 0 to 2.0, and the drug release decreased with lower molecular weights of HPC according to the investigation of Johnson et al. [21]. 

More importantly, the selection of the test conditions is crucial for the in vitro–in vivo correlation (IVIVC) [14,26,27]. Scheubel et al. showed that using the IVIVC with HCl solution to predict the performance of a drug in vivo was superior to using other pH values and types of media [28]. Additionally, for phosphate buffer solutions with different buffer capacities and ionic strengths, the release profiles and IVIVCs were significantly different, which revealed that an inappropriate in vitro model would lead to low predictions [18]. Generally, it is necessary to have a deeper understanding of in vitro processes and the influences of the EC/HPC film to provide better insight for polymer-coated pellets. However, there is little reporting on the effect of the release media on the drug-release kinetics and microstructure of the EC/HPC film, as well as the release mechanism.

In this article, metoprolol succinate (MS) was used as the model drug due to its good aqueous solubility irrespective of the pH of the media, a relatively short plasma half-life (*t*_1/2_ = 3–4 h) and accuracy and simplicity of analytical method (by ultraviolet spectrophotometer) [14,29,30]. The different blend ratios of EC/HPC (100:0; 80:20, 78:22, 76:24, 74:26 *w*/*w*) coated MS-pellets and the corresponding free-coating films obtained by spraying were designed and prepared. The in vitro drug release characteristics of the pellets were evaluated in deionized water, HCl solution at pH 1.2, acetate buffer at pH 4.5 and phosphate buffer at pH 6.8, respectively. The influence of the media (pH, compositions, and osmolality/ionic strength) on the drug release, polymer composition, water uptake, microstructure and mechanisms of mass transport was also studied. In addition, a mathematical theory was used to better explain and understand the polymer distribution and drug release mechanism conversion in different media for EC/HPC films. The deeper insights obtained into how the film coating performs controlled drug release in media will benefit the design and development of pharmaceutical EC/HPC systems.

## 2. Materials and Methods

### 2.1. Materials

Ethyl cellulous (EC, ethocel standard 10 premiums, Colorcon Coating Technique Company, Shanghai, China). Hydroxypropyl cellulose (HPC grade LF, Hercules Incorporated, Wilmintong, IL, USA). MS Celphere (CP305, Asahi Kasei Chemicals Corporation, Tokyo, Japan). Metoprolol succinate (99.5% purity, Jiashi Lianbo Medical Development, Co., Ltd., Beijing, China). Sodium chloride (NaCl, Sinopharm Chemical Reagent Co., Ltd., Beijing, China). Ethanol (95%, Beijing Zhenyu Minsheng Pharmaceutical, Co., Ltd., Beijing, China) was used for polymer dissolution. Hydrochloric acid, sodiumhydroxide, potassium dihydrogen phosphate, sodium acetate, acetic acid, potassium chloride and potassium nitrate were purchased from Sinopharm Chemical Reagent, Co., Ltd. (Shanghai, China). All reagents were of either analytical or chromatographic grades.

### 2.2. Solubility of Metoprolol Succinate

The solubility of the metoprolol succinate was measured by loading over-dose drug into 10-mL penicillin bottles containing 5–8 mL different release media (*n* = 3), which were shaken for 72 h at 37 °C in a shaking table (HNY-200B, INCUBATOR SHAKER, Tianjin, China). The saturated drug solutions were diluted and extracted through a 0.45-µm filter, then assayed by a UV spectrophotometer (UV-1750, Shimadzu, Japan) after appropriate dilutions (λ = 274 nm; UV-1750). The plateau values were considered as the drug solubility. 

### 2.3. Preparation of Coated Pellets 

#### 2.3.1. Drug-Layered Starter Cores

Pellet starter cores were coated with a drug solution consisting of 60% (*w*/*v*) metoprolol succinate, deionized water as solvent in a fluidized bed coater (GPCG-1, Glatt GmbH, Binzen, Germany). The process parameters were as follows: inlet air temperature 65 °C, spray rate 10–13 g/min, atomization pressure 2.0 bar. The final drug loading was 0.6 g/g which was determined by UV spectrophotometry at a wavelength of 274 nm (UV-1750, Shimadzu, Japan), and the layered MS-MCC cores were sieved through 40-mesh and 30-mesh sieves to remove undersized or oversized particles after the layering process.

#### 2.3.2. Film-Coated Controlled Release Pellets

The drug-layered MCC cores were coated with different blend ratios of EC/HPC (100:0, 80:20, 78:22, 76:24, 74:26) solution in 95% *v*/*v* ethanol in a GPCG-1 fluidized bed coater using the bottom spray method (GPCG-1, Glatt GmbH, Binzen, Germany). The polymer solution (polymer comment = 6% *w*/*w*) was placed under vigorous magnetic stirring overnight. The coating conditions were as follows: inlet air temperature 55 ± 3 °C, atomization pressure 2.0 bar, spray rate 12.4–16.6 g/min. After coating, the pellets were further fluidized at 60 °C in the fluidized bed for additional 20 min to reduce the residual solvents. The average thickness of the film was 42.15 ± 5.29 µm. 

### 2.4. Characterization of Coated Pellets

#### 2.4.1. In Vitro Drug Release Measurement

The in vitro drug release experiments were performed under sink conditions upon exposure to 500 mL deionized water, HCl solution at pH 1.2, phosphate buffer pH at 6.8 and acetate buffer pH at 4.5 using the USP 35 paddle method at 37 °C, respectively. Stirring rate was 100 rpm (RCZ-8M dissolution tester, Tianjin, China). 5 mL samples were withdrawn at predetermined time points (1, 2, 4, 8, 12, 16, 20, 24 h), then filtered through a 0.45-µm filter, and using a UV-spectrophotometer to analyze the content of drug (λ = 274 nm, UV-1750, Shimadzu, Japan). All drug release studies were conducted in triplicate.

#### 2.4.2. Pellet Swelling Monitoring

To study the pellet swelling behavior, the size of the pellets during dissolution was monitored. At predetermined time points, we carefully removed the excess water on the surface of sample pellets, and the diameter of pellets was measured in the wet state (diameter _(t)_) using an optical microscope analysis system (Nikon ECLIPSE E-100, Tokyo, Japan) equipped with a Nikon camera (Nikon D5100, Tokyo, Japan). The increase in pellet diameter (%) at each time was calculated as
(1)$${\mathrm{increase}\;\mathrm{in}\;\mathrm{pellet}\;\mathrm{diameter}\;(\%)}_{\left(t\right)}=\frac{{\mathrm{diameter}}_{\left(t\right)}{-\mathrm{diameter}}_{\left(0\right)}}{{\mathrm{diameter}}_{\left(0\right)}}\cdot 100\%$$
where diameter _(0)_ is the diameter of pellets at time *t* = 0. Each dissolution cup contained 100 pellets upon exposure to each medium

#### 2.4.3. Osmotic Pressure Determination

To obtain the effects of osmotic pressure in the different media on drug release from the coated controlled-release pellets, NaCl was added to adjust the pressure of the media (deionized water, HCl solution at pH 1.2, phosphate buffer at pH 6.8 and acetate buffer at pH 4.5). The osmolality of the release media and metoprolol succinate aqueous solutions were performed with an osmometer (SMC 30C, Tianhe medical instrument Co., Ltd., Tianjin, China), which uses the freezing point depression method [6,31]. Release studies were performed as described in Section 2.4.1. Each experiment was carried out in triplicate. 

### 2.5. HPC Leaching and Viscosity in Different Media

To simulate the release of the HPC pore-former from the controlled-release system, we used the coated pellets to determine the HPC leakage profiles. The samples were analyzed by size exclusion chromatography and refractive index detection (SEC-RI). The reference substance was dissolved in a mobile phase containing 10 mM NaCl and 0.02% *w*/*v* NaN_3_ for at least 48 h prior to analysis. The column was a TSK gel GMPWXL 7.8 mm D × 30.0 cm L, with a mean particle size of 13 μm (TOSOH Corporation, Japan). The sample concentration was 0.2 mg/mL, and the volume of the injected sample was 100 µL using a flow rate of 0.5 mL/min. 

The viscosity of polymer in solution depends upon the physicochemical property of it, such as the chemical structure, molecular weight and its interactions with the solvent [16]. Formulating a range of concentrations of water-soluble HPC solutions in the different study media (*n* = 3, 37 °C), and determination of the viscosity of the solutions were achieved with a viscometer (Fungilab, VISCOLEAD ADV, Co. Ltd., Barcelona, Spain). Each experiment was conducted in triplicate.

### 2.6. Preparation of Free Films

The use of free films to predict coating properties and mass transport has been reported as a convenient method [5]. Therefore, EC/HPC free films were prepared by spraying a solution of 6% *w*/*w* polymer and 94% *w*/*w* ethanol (95% *v*/*v*) through a moving atomizer nozzle onto a rotating cylinder (diameter = 60 mm, length = 80 mm) in a controlled temperature and air flow. The setup was designed to simulate fluidized bed for film spraying on pellets, which has been reported elsewhere [1,15]. The cylinder was first covered with 70 mm wide Teflon tape to help film removal from the cylinder after drying. The spraying process parameters were as follows: inlet air temperature 70 ± 2 °C, atomization pressure 2.0 bar, spray rate 12.45 g/min, cylinder rotation 80 rpm. After spraying, the film was left to dry for an additional 20 min. For each film, the amount of polymer dry weight was about 6 g, giving it a final thickness of 90–110 μm, which ensured sufficient detection for SEM analysis. The films were stored at room temperature in a desiccator containing silica gel.

### 2.7. Water Uptake and Microstructure of Free Films

The water-uptake kinetics of the free films were determined by gravimetrically upon exposure to deionized water and different concentrations of acetate buffer with a pH of 4.5, optionally containing 80% *w*/*w* EC and 20% *w*/*w* HPC. Pieces of 2 × 2 cm^2^ were placed into a 300-mL dissolution cup filled with 200 mL preheated media (37 °C, 100 rpm, ZRS-6G, dissolution tester, Tianjin, China; *n* = 3). At predetermined time points (0.25, 0.5, 0.75, 1, 2, 4, 8, 24 h), the films were withdrawn and were accurately weighed (wet mass _(t)_) after the excess surface water was carefully removed, then the film was dried to a constant mass at 60 °C (dry mass _(t)_). The water content (%) was calculated as
(2)$$\mathrm{water}\;\mathrm{content}\;{\left(\%\right)}_{\left(t\right)}=\frac{{\mathrm{wet}\;\mathrm{mass}}_{\left(t\right)}{-\mathrm{dry}\;\mathrm{mass}}_{\left(0\right)}}{{\mathrm{wet}\;\mathrm{mass}}_{\left(t\right)}}\cdot 100\%$$

The microstructures of the different blend ratios of EC/HPC films before and after exposure to the release media were studied using a scanning electron microscope/SEM (S-4800 Analytical Electron Microscope, Hitachi, Japan). Film pieces, approximately 2 × 2 cm^2^ each, were added to a 300-mL dissolution cup filled with 200 mL medium and incubated for 24 h at 37 °C (100 rpm, ZRS-6G, dissolution tester, Tianjin, China; *n* = 3) to certify that both sides of the pieces were exposed to the media. After 24 h, the films were dried in a desiccator containing silica gel for ≥ 3 days to test. The samples were loaded onto a copper sample holder, sputter-coated with carbon followed by gold in a vacuum using a sputter coater (E-1010, Hitachi, Tokyo, Japan), and then observed under an excitation voltage.

### 2.8. Mathematical Modeling

The type of pellets considered in this study consisted of a solid core coated with a polymer film. The total mass balance for this type of coated formulation upon exposure to an aqueous solution can be written as [5]
(3)$$\frac{dM}{dt}=\rho \cdot Jv$$
where *M* is the mass of the coated formulation, *ρ* is the density of the bulk. *Jv* is the net volumetric bulk flow through the coating and depends upon the flow of both the medium and the drug. 

For a homogeneous film, *Jv* also can be written according to the reversible thermodynamic equation
(4)$$Jv=\frac{K}{h}\cdot A\cdot (\sigma \cdot \Delta \pi -\Delta P)$$
where *K* is the water permeability of the film with respect to the solvent, *σ* is the drug reflection coefficient, *h* is the coating film thickness, and *A* is the surface area of the coating film. The osmotic pressure difference across the film is Δ*π*, and Δ*P* stands for the hydrostatic pressure difference between the two sides of the film. The reflection coefficient *σ* is a correction factor that describes the sieving capability of the coating film for the solute [32]. When the film is completely impermeable, it only allows water to flow through the pores/cracks and not the solute molecules. In that case, the hydrostatic pressure will be built up inside the pellets and the reflection coefficient is 1. For a semipermeable or leaky film, it is similar to impermeable film, the reflection coefficient was between zero and one, and the release mechanism is mainly osmotic pumping when 0 < *σ* ≤ 1. When the pore sizes in film become larger than the diameter of the solute molecule, allowing the solute outflow, the drug reflection coefficient approaches 0, which causing the osmotic pressure to be negligible and the drug release to occur mainly by diffusion. Furthermore, the Δ*P* in Equation (4) can be replaced by Δ*C*, which means the drug concentration difference across the permeable film, to represent the drug release rate from the coated formulation. The total mass balance through a coated formulation upon exposure to any aqueous solution can be written as
(5)$$\frac{dM}{dt}=\frac{A}{h}\cdot K\cdot (\sigma \cdot \Delta \pi -\Delta P)\cdot C_{s}+\frac{A}{h}\cdot D\cdot \Delta C$$
where *C_s_* stands for the drug concentration of the saturated solution and *D* is the diffusion coefficient of the drug through the coating polymer films.

## 3. Results and Discussion

### 3.1. Drug Release from EC/HPC Film-Coated Pellets 

#### 3.1.1. Effects of the Media pH 

The profiles of drug release from the EC/HPC coated pellets (the HPC level was 20%, 22%, 24% and 26% *w*/*w*, respectively) were measured in deionized water, HCl solution at pH 1.2, phosphate buffer at pH 6.8 and acetate buffer at pH 4.5, respectively, and the results are shown in Figure 2. As expected, HPC acted as pore-former in the coat only within a small range from 20% to 26% *w*/*w* of the coating and could regulate the release of drug from the coated pellets to a great degree. Interestingly, some unusual tendencies were also observed as follows. The release of the drug was faster in acetate buffer at pH 4.5 and slower in phosphate buffer at pH 6.8 than that in the other media, regardless of the level of HPC. However, when the release medium was HCl solution at pH 1.2, the tendency and acceleration of drug release were dependent on the HPC levels. As the HPC increased, the drug release rate changed from slow (20%) to fast (24%) compared with that in the other media. The solubility of the metoprolol succinate in the respective media at 37 °C (Table 1) confirmed that although the solubility of drug tends to decrease with increasing pH values (except for water), the differences is relatively small and cannot be fully attributed to the different release profiles. Therefore, it could be concluded that the different drug release rates are may influenced by the effects of the media on the coating polymer. 

As previously shown, the drug release mechanism is primarily controlled by osmotic pumping in 20% HPC films (in which the drug release rates from a semipermeable film will decrease with the osmotic pressure of the release medium increases) and by diffusion with 24% HPC films (in which the drug release rates will increase as the concentration gradient across the porous film increases), respectively [3,5,6]. Therefore, with a lower percentage of HPC, the slower drug release observed in pH 1.2 and 6.8 buffer than that in water can be attributed to the higher osmolarity of the bulk solution; however, this was not consistent with the drug release in acetate buffer, in which the osmolarity was more than that in water (Table 1). Additionally, the drug burst release in HCl solution with a higher percentage of HPC cannot be explained by the limited drug solubility alone. Therefore, this suggested that the different release media likely played an important role in changing the microstructure of the polymer films (e.g., the film permeability and sizes of the pores or channels) by affecting the properties of EC or HPC, thereby altering the release mechanisms of the coated pellet. 

#### 3.1.2. Effects of Media Composition

Based on the unusual drug release rates from the pellets in HCl solution at pH 1.2 and acetate buffer at pH 4.5, the release rates of the drug were further studied in different ionic compositions and strengths of buffers to evaluate the media effects. The release profiles obtained are illustrated in Figure 3. In one profile, when the pore-former level was 24%, the drug release rate decreased as the value of pH in the HCl solution increased from 1.2 to 3.0 (ionic strength decreased from 0.06 to 0.001), but there were no differences in the release rates when a 0.1 M solution of potassium chloride or a 0.1 M solution of potassium nitrate was added into the HCl solution at pH 1.2 (Figure 3A), which means that the fast drug release in HCl solution was mainly attributable to the concentration of hydrogen ions (above 0.001 M), rather than to the anion of the media. Meanwhile, with the concentration of acetate buffer increased, the drug solubility and release rate also increased, but the differences in rates were not significantly at the range of 0.05–0.5 M. In the other profile, when the pore-former level was 20%, the release rates of the drug were more rapid in acetate buffer at pH 4.5 than in phosphate buffer at pH 4.5, and interestingly, the opposite effect was observed for the acetate buffer concentration (the effects of the ionic strength of the medium) on the resulting drug-release kinetics from semipermeable films. Upon increasing the concentration of the acetate buffer (pH 4.5), the release rate of the drug significantly increased compared with the release rate from 24% HPC pellets (Figure 3B), which revealed that the mass transport from the EC/HPC film was altered in the presence of these carboxyl groups and the higher the ratio of EC in coating, the greater the influence on drug release, regardless of the osmolarity of the buffer. In contrast, upon exposure to the phosphate buffer and HCl solution, the release rate was only dependent upon the osmolarity rather than the pH value, which was in accord with the osmotic mechanism for a leaky film (the data measured in HCl solution are not shown). These results indicated that variations in the media composition will influence the underlying drug release mechanisms and might bring about unexpected changes in the final drug release patterns, which should be taken into account during such product design and optimization. 

### 3.2. Microstructure of Free EC/HPC Film

To clearly understand the reasons for the above-described tendencies, SEM observations of free films in different media were conducted (Figure 4 and Figure 5). Interestingly, the microstructure of EC/HPC films was in consistent with the results of the drug release from pellets characteristics measured in the tested media. First, no remarkable cracks or pores were observed in the films before the experiments, irrespective of the level of HPC (Figure 4A and Figure 5A). The white solid on the surface may be product leftover from the coating solution and did not affect the experimental results. To summarize, the SEM images respectively represented by films containing 20% and 24% HPC showed several phenomena. (i) As the amount of HPC increased, the number of self-formed pores in the films increased, thus increasing the porosity and permeability of the EC/HPC films and accelerating the drug release (Figure 2, Figure 4 and Figure 5). (ii) With 20% HPC in the free films, the pore sizes were strongly affected by the acetate buffer (Figure 4), producing deeper and larger pore sizes than those in the films facing the other media, especially as the concentration increased, which caused the sizes of the pores in the films to become larger (0.18 ± 0.16 µm, 0.21 ± 0.19 µm, 0.35 ± 0.17 µm, and 0.52 ± 0.31 µm measured in deionized water and 0.005 M, 0.05 M, and 0.5 M acetate buffers, respectively). (iii) Furthermore, films with 24% HPC immersed in HCl solution at pH 1.2 showed the largest pore size (0.73 ± 0.43 μm) compared to acetate buffer at pH 4.5 (0.05 M, 0.51 ± 0.19 μm) and water/pH 6.8 phosphate buffer (0.22 ± 0.11 μm), producing films which could diffuse drugs more easily (Figure 5). 

In conclusion, based on the results of drug release and microstructure testing of the free films, it was concluded that the type and ionic strength of the media largely affected the pore structure of the coating films. Especially, films containing different levels of HPC will be significantly controlled by acetate buffer and HCl solution, respectively. The pore shape, in turn, can be related to film permeability and pore connectivity, which play important roles in controlling the drug release and release mechanism [33,34]. Therefore, our study added the evidence to explain the different mass transport processes in that the structure of the film is greatly influenced by the polymer composition and pore-former levels, release media in particular. 

### 3.3. Influence of Media on HPC Properties

HPC is a hydrophilic polymer which can be hydrated, swollen and solubilized in water to act as a pore-former, forming pores or channels in the coating film or making the polymer film microporous and inducing drug release [24]. The major factors governing the dissolution rate, hydration and gelation or erosion rate of the HPC are the chemistry, viscosity, concentration, and particle size of the HPC, which will influence the drug release from the polymer films to a certain extent [35,36]. Normally, the drug release rate increases with increased leaching of HPC and the more viscous the hydrophilic polymers, the faster is the swelling of its side chains and the greater is the resistance to polymer erosion, slowing the drug release rate [5,16,37,38,39].

#### 3.3.1. HPC Leaching from Coated Pellet 

Figure 6 shows the HPC leaching profiles from the film-coated pellets with different levels of HPC in different media. First, less HPC (4%) was leached from the coating films containing 20% HPC after soaking for 24 h in deionized water. However, when the HPC level increased from 22% to 26%, the dissolution of HPC increased drastically (from 27% to 65%) (Figure 6A). In this case, HPC is probably capsulated by EC when the level is below 22%, then the leaching is thereby hindered [5,40]. When the level is high, a continuous phase rich with HPC will be created, and pores or channels can be formed easily throughout the film due to the quick leaching of the HPC-rich phase, accelerating the release of the drug [1]. Nevertheless, although the water-soluble HPC mainly controls drug release, there were no fundamental differences in the cumulative amount of HPC leached between the different media, regardless of the HPC level or the type and ionic strength of the media, surprisingly (Figure 6B,C). In comparison with the related drug release profiles, when one unit of HPC was dissolved, the amount of drug released from the pellets in HCl solution or acetate buffer was about twice that in water. This further indicated that the pore sizes of the films became larger and better able to diffuse more of the drug across the coating film, rather than this observation being the effect of the amount of HPC dissolved. Therefore, the other properties of HPC changed in different media might act as important factors affecting the pore structure of free films and the overall release of a drug.

#### 3.3.2. Viscosity of HPC Solution

Figure 7 shows the differences in viscosity of the HPC solution between HCl solution at pH 1.2, acetate buffer at pH 4.5, phosphate buffer at pH 6.8 and deionized water, respectively (the results in water as contrast). The concentration of HPC in the solution was 0.5, 1.0, 2.0, 4.0, 6.0, 8.0, and 10.0 *w*/*w*. Initially, there were no differences in viscosity among the media with low concentrations of HPC, this could be owing to the HPC is dispersed as a very small region. However, as the amount of HPC increased, diversity appeared. As the concentration of HPC in the solution increased, the decline in viscosity in HCl solution compared to water was more than that in acetate buffer and phosphate buffer. It can be concluded that the percentage of erosion increased and the percentage of swelling decreased as the viscosity of HPC decreased, forming a weak gel layer upon the entry of HCl solution into the polymer, which may also weaken the binding of EC, thus changing the microstructure of the coating by increasing the size of the pores formed in the films (Figure 5C). Meanwhile, as the viscosity and swelling of the gel layer decreased, the thickness of this region became thinner and the distance that the drug had to travel to be diffused decreased, which increased the drug release rate. Therefore, although the amount of HPC leached from the pellets was similar in the different media, the viscosity of HPC changed upon exposure to the HCl solution, may occur and influence the nature, permeability and microstructure of the coating, thus influencing the drug release from EC/HPC system. 

Likewise, based on the Equation (5), when the film became permeable to the drug, the reflection coefficient *σ* approached 0, and the drug could be diffused from the larger pores in the coating films, producing a positive correlation between the drug release rate and the drug solubility. In this case, the faster release in HCl solution can also be attributed to the greater solubility: 303.53 vs. 276.05 mg/mL (at 37 °C) at pH 1.2 and in water, respectively. In conclusion, when the level of HPC is above 22% *w*/*w*, the drug release rates from the film depend on both the effect of the concentration of hydrogen ions (pH) on the HPC-rich area and drug solubility. 

### 3.4. Influence of Media on EC-Coating Film Properties

#### 3.4.1. Drug Release From Pellets 

Pure EC-coated pellet release experiments were performed to investigate the possible effect of the media on the drug-release kinetics and microstructure of the polymer film. The pure-EC pellet release results are presented in Figure 8. It can be seen that all the drug release kinetics are consistent with the results from pellets containing 20% HPC. In general, EC is a water-insoluble polymer regarded as poorly permeable to most drugs and that follows the principle of osmosis. However, the higher drug release rate in the higher osmolality of the acetate buffer indeed showed that the presence of acetate/acetic acid in the medium could influence the permeability and pore structure of the EC film. Thus, comprehensive understanding of the release mechanism from EC-coated pellets during the whole mass transport process may be more necessary. The total zero-order drug release rates (R_T_) from EC/HPC (100:0, 80:20) pellets in each medium were estimated from the linear portion of the release curves (*R*^2^ > 0.99), and the data are summarized in Table 2. As it can be seen, with the increase in the concentration of acetate buffer at pH 4.5, the increase of the R_T (pure EC)_ component was more significant than that of the R_T (80% EC)_ component of the release rate, as indicated by the ratio of R_T (pure EC)_ to R_T (80% EC)_; however, with the amount of HPC increased, the overall drug release rate increased, decreasing the contribution of pure EC to the release rate (0.29 vs. 0.20 vs. 0.37 measured in 0.05 M acetate buffer with 22% HPC, 0.05 M and 0.5 M acetate buffer with 24% HPC, respectively). Hence, it was further proven that the reason for the unusual drug release rate in acetate buffer were might mainly depend on the higher concentration of acetate buffer alter the properties and permeability of the EC-rich area (>78%). Additionally, the drug solubility was slightly increased when the concentration of acetate buffer was within 0.005 M–0.5 M (259.93 mg/mL vs. 304.79 mg/mL, Table 1), facilitating dissolution to a certain extent according to the Noyes–Whitney equation. 

#### 3.4.2. Water Uptake and Swelling Study 

The water permeability of free films (100 µm) and the swelling kinetics of the pellets were determined in different concentrations of acetate buffer at pH 4.5 and 37 °C, respectively (Figure 9). As expected, there was an accordant trend between the water uptake kinetics of the polymer films, the swelling kinetics of the pellets and the drug release profile. First, the water content was positively correlated with the concentrations of acetate buffer, resulting in an increase in the permeability of the films (Figure 9A). Likewise, Figure 9B shows the changes in the diameter of the coated pellets, which substantially increased in the first 2 h in all three media; however, as the concentration of the acetate buffer was reduced, there was significantly prolonged time point during which the diameters of the pellets decreased. This is because (i) initially, the coating film is semipermeable to the drug, only allow water driving in, producing dissolution of the drug. (ii) Because of the mass accumulation inside the core, swelling of the pellet. (iii) Then, due to the dissolved drug outflow through the cracks or small pores in coating film, caused the pellet size to decrease. Based on the results from the water content and swelling experiments, the higher the concentration of the acetate buffer, the faster is the water absorption of the film and the swelling rate of the pellets, resulting in an increased drug dissolution rate and shortening the time to maintain saturation. As a result, the release of the drug became increasingly fast. Furthermore, the pellet variability in the media at 37 °C after 6 h is also reflected in the microscope pictures in Figure 10. As it can be seen, almost all pellets decreased in size because of the drug being released from the pellets after 6 h (Figure 9B). Some looked like deflated balloons. The opaque black contents of the pellets can be seen as the undissolved MCC core and the drug. The nonopaque contents of the pellets can be explained by the dissolved drug solution and the increasing concentration of acetate buffer at pH 4.5. The increase of nonopaque contents of the pellets further demonstrated that the acetate buffer concentration can increase the hydrophilicity of the EC films and, thus, affect drug release behavior (the ratio of nonopaque pellets over the total pellets after 6 h: 0.005 M-2.5%; 0.05 M-6%; 0.5 M-30%, *n* = 100).

The results obtained from the swelling kinetics of the coated pellets and water uptakes of the free films can, thus, help in better understanding the unexpected drug release profiles observed in coated pellets containing below 20% HPC in the acetate buffer at pH 4.5. The differences in the drug release behavior could be mainly attributed to the effects of the EC interacting with the acetate buffer, rather than to the osmolality, resulting in the membrane becoming more transparent and more water permeable and changing the overall drug release mechanism from zero-order release to first-order release.

#### 3.4.3. Microstructure

The surfaces of the pure-EC films displayed great differences in the different media, especially in acetate buffer (Figure 11). Surprisingly, the images clearly show that some small pores appeared in the surface of the pure-EC film after being submerged in acetate buffer at pH 4.5 but were not found in other media. The higher concentrations of acetate buffer, the more and larger pores are formed in films (none, 0.041 ± 0.033 µm, and 0.061 ± 0.029 µm measured in water and 0.005 M acetate buffer, 0.05 M, and 0.5 M acetate buffer, respectively). The drug released from the coated pellet followed by the large-pore transport will be the key factors for fast drug release. The results were in consistent with the water permeability of the film and the drug release from the pellets. Overall, when immersed in the release media, the solvent may penetrates into the free spaces between macromolecular chains of the polymer to increase its mobility, and the water solubility of a polymer is related to its ability to establish hydrogen bridges between the carboxyl groups of the acetate buffer and the hydroxyl groups of the polymer, such that the higher the number of strong polar carboxyl groups is, the faster the hydration is [16,41]. The dimensions of the polymer molecule may also increase due to polymer relaxation from the stress of the penetrated solvent, which could increase the porosity, permeability, and hydrophilicity of the film. Therefore, the final size of the membrane pores may depend on the combined action of EC and HPC, rather than on the impact of HPC alone.

### 3.5. Release Mechanism

To better understand and verify the validity, reliability and extent of the roles of above-described unusual tendencies of the release mechanism (osmosis/diffusion), the release of metoprolol succinate from EC/HPC (80:20, 78:22, 76:24 *w*/*w*) coated pellets was further investigated in different osmotic potential release media by adding adequate amounts of NaCl (deionized water, HCl solution at pH 1.2, acetate buffer at pH 4.5, and phosphate buffer at pH 6.8) (Figure 12). The zero-order release rates were estimated from the linear portions of the release profiles (*R*^2^ > 0.98). Clearly, increasing the osmolality of the release media caused a decreased in the rate of release, irrespective of the HPC levels and the type of media, indicating that the release of drug from EC/HPC-coated pellets will be affected by osmotic mechanisms, but the extent will differ. Interestingly, the decrease of the drug release rates with increased amounts of NaCl in the media was opposite to the results for the increased osmolality of the acetate buffer and the HCl solution itself (Figure 3). With same osmolality of the media, the drug release rate was also faster in the acetate buffer and HCl solution than in the other media when the pellets contained 20% *w*/*w* and 24% *w*/*w* HPC in the coating, respectively. The results further certified that the amount of acetate buffer and HCl can influence the structure of the EC/HPC film and the form of mass transport from the pellets.

Theeuwes and Higuchi described the drug total zero-order release rate from a coated formulation to be the sum of the contributions from the osmotic pumping rate and transmembrane diffusive flux of the drug, using the following relationship [42]
(6)$$R_{T}=R_{O}+R_{D}=\frac{A}{h}\cdot K\cdot \Delta \pi \cdot C_{S}+\frac{A}{h}\cdot D\cdot C_{S}$$
where the Equation (6) is transformed from Equation (5), in which Δ*P* is typically negligibly small compared to Δ*π*, and the Δ*C* can be replaced by *C_s_* because the drug concentration in the external media is very low compared to that inside the pellets [43]. Therefore, R_O_ is the contribution from the osmotic mechanism to the zero-order release rate of drug, and R_D_ is the contribution from the diffusion mechanism to the zero-order release rate of drug. According to Equation (6), the slope (k) of the linear function indicates the capability of water and solution to influx into the film under a pressure gradient, and the intercept indicates the contribution from the diffusive flux to zero-order release rate (R_D_)_._

The osmolality of the external medium and saturated drug aqueous solutions determined by the freezing point depression method are summarized in Table 1. When the osmolality in the pellet core was equal to the osmolality of the saturated drug aqueous solution, a linear regression was performed on the total drug zero-order release rate R_T_ vs. the osmolality difference between the core and the external medium Δ*π*, which data are presented in Figure 13, and the regression parameters R_T_, R_O_, and R_D_, as well as R_O_ over R_T_ values are also summarized in Table 3.

The positive correlation between the R_T_ and Δ*π* confirms that the osmosis-mediated drug release mechanism was involved in the release of all media from the coated pellets. From Table 3, the slope increased as the amount of HPC increased when fixed in each medium, which indicated that HPC leakage increased the number of self-formed pores and produced higher water permeability through the membrane. However, with a fixed blend ratio of HPC (22% and 24%), the slope was reduced in solution at pH 1.2 and pH 4.5, which indicates a decrease in the water penetration rate into the system by osmosis and an increase in the drug diffusion from the large-diameter pores (Figure 5), which weakens the osmosis-driven effects. Therefore, the effects of the release media on these parameters can be quantified, a deeper and comprehensive insight into the underlying drug release mechanisms can be gained.

#### 3.5.1. Deionized Water and Phosphate Buffer at pH 6.8

The results of the linear function determined with different ratios of HPC-coated pellets upon exposure to deionized water and phosphate buffer at pH 6.8 are shown in Figure 13A,B, respectively. As can be seen, despite the increasing amount of HPC, the release occurred mainly by osmotic pressure, and most of the coating was still semipermeable (Table 3). When the films contained 20% HPC, the release mechanism was totally due to osmotic pressure and the hydrostatic pressure (Δ*P*) generated inside the core (see Equation (4)); however, as the leaching of HPC increased, the extent of the osmolality-driven release slightly decreased, and the hydrostatic pressure disappeared, as indicated by the ratio of R_O_ over R_T_. Additionally, note that there was little intercept in pellets with 22% and 24% HPC, indicating a minor contribution from drug diffusion through the polymer. This phenomenon can be explained by the formation of pores in 20–24% HPC films immersed in water and phosphate buffer at pH 6.8 which allow the influx of water are smaller than the sizes of individual drug molecules, and the nature of the polymer is stable in these media (Figure 4B,D and Figure 5B,D). Therefore, although the amount of HPC and pores increased, the size of the pores did not increase remarkably. As a result, the drug release from the pellets coated with 20%-24% HPC in water and phosphate buffer at pH 6.8 is mainly driven by osmotic pressure, with a minor contribution from diffusion.

#### 3.5.2. HCl Solution at pH 1.2

The total zero-order release rates from the coated pellets in HCl solution at pH 1.2 obviously increased as the ratio of HPC increased from 20% to 22%/24% (Figure 13C). Meanwhile, with an increased amount of HPC, the increase in the R_D_ component was more significant than that of the R_O_ component of the drug release rate, as indicated by the ratio of R_O_ over R_T_ summarized in Table 3. The release mechanism transformed from totally osmotic to mainly by diffusion, which can be owing to the self-formed pores of a large size for drug solution efflux (Figure 5C). In a word, this mathematical modeling further verified that the pore-structure of the coating film will be significantly responsible for differences in drug release rates for the medium of HCl solution at pH 1.2. With the pore sizes in film became larger than the solute molecule’s diameter, the film’s sieving capability was lost, causing the osmotic component to be negligible. 

#### 3.5.3. Acetate Buffer at pH 4.5

The results of the linear function determined from the exposure of different ratios of HPC-coated pellets to acetate buffer at pH 4.5 were consistent with our findings concerning the pore structure of EC/HPC films (Figure 13D). When the pellets were coated with 20% HPC, although the osmolality of the acetate buffer at pH 4.5 was higher than that of water (70 vs. 0 mOsmol/kg), the drug release rate was still faster in acetate buffer. From the values of R_O_ over R_T_ shown in Table 3, the mechanism is always a combination of osmotic and diffusion effects, regardless of the level of HPC, which means that the pore sizes became larger than that in the other media, which is proven in Figure 4. When the level of HPC was increased from 22% to 24%, the contribution of R_O_ to the zero-order rate exhibited a steady trend, demonstrating that the pore size of the coating is not mainly dependent on the EC but also on the amount of HPC. Overall, the drug release pattern and mechanism can be influenced by the acetate buffer, especially affecting the properties of high-dose polymers of EC to obtain the faster release of drugs. 

In summary, the results of the osmotic pumping and diffusion mechanisms and both together in different media could better explain the abovementioned unusual drug release profiles. The property/structure of the coating film play an important role in influencing the drug release mechanisms. The larger the size of the pores in the films, the higher is the permeability of film and the contribution of diffusion mechanism to accelerate the release rate, and the more drugs diffuse through the coated pellets. It is clear that the diffusion mechanism is mainly attributed to the larger variation in pore size when the pellets are upon exposure to different media, while the smaller pores only participate in the osmosis-mediated flux of drugs. Therefore, study of the mechanisms reconfirmed that the release media can affect the pore structure of the EC/HPC films, affecting the mass transport properties of the final film.

## 4. Conclusions

The drug release mechanism and microstructure of EC/HPC film-coated pellets were affected by the film composition and the release media. Unexpected drug release patterns were observed in HCl solution at pH 1.2 and acetate buffer at pH 4.5, which may be respectively attributed to the change in the property of the HPC-rich area (>22%) and EC-rich area (>78%) rather than in the pH-independent dissolution of HPC and solubility of the drug in the media. As a result, the pore size and permeability of the polymer films were significantly increased, making the diffusion mechanism become the dominant model based on the SEM and R_O_/R_T_ results, thus obtaining a fast drug release profile. Our study indicated the range of how drug release is affected by media and provided a mechanistic understanding of how this type of drug delivery system controls drug release, which can be very helpful in the design and optimization of ideal film-coated formulations.

## Figures and Tables

**Figure 1 pharmaceutics-11-00080-f001:**
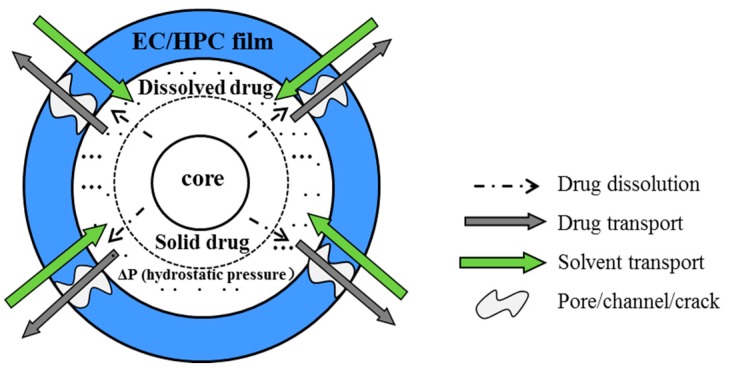
Schematic illustration of the drug release process from a film-coated pellet.

**Figure 2 pharmaceutics-11-00080-f002:**
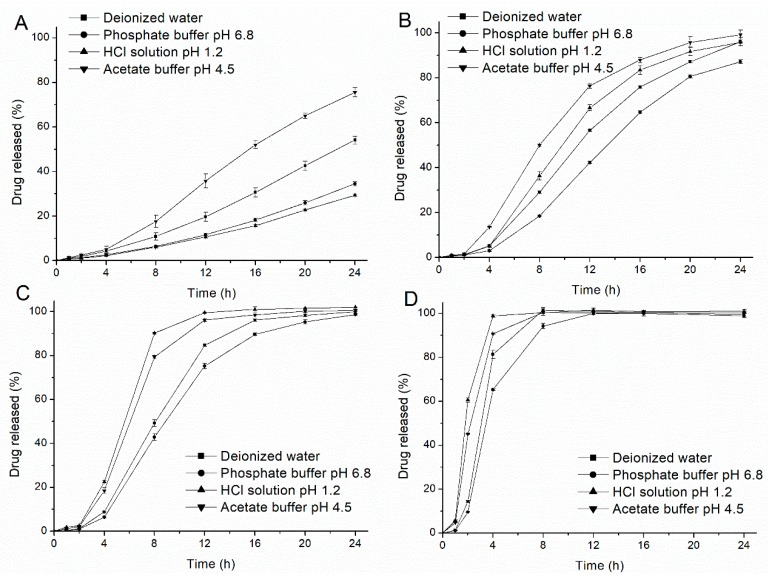
Drug release profiles from coated pellets with different blend ratio of ethyl cellulose (EC) and hydroxypropyl cellulose (HPC) in deionized water, HCl solution at pH 1.2, acetate buffer at pH 4.5 (0.05 M) and phosphate buffer at pH 6.8 at 37 °C, respectively. (**A**) EC:HPC=80:20; (**B**) EC:HPC=78:22; (**C**) EC:HPC=76:24; and (**D**) EC:HPC=74:26. Error bars represent standard deviation.

**Figure 3 pharmaceutics-11-00080-f003:**
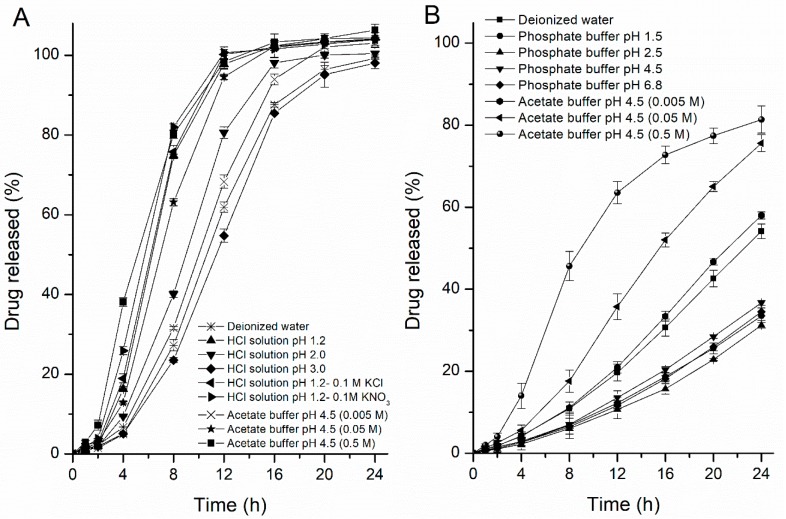
Drug release profiles from pellets in different media. (**A**) Influence of different anions, pH values of the HCl solution and concentration of the acetate buffer at pH 4.5: pellets containing 24% *w*/*w* HPC. (**B**) Influence of different pH values, compositions and ionic strengths of the release media: pellets containing 20% *w*/*w* HPC. Error bars represent standard deviation.

**Figure 4 pharmaceutics-11-00080-f004:**
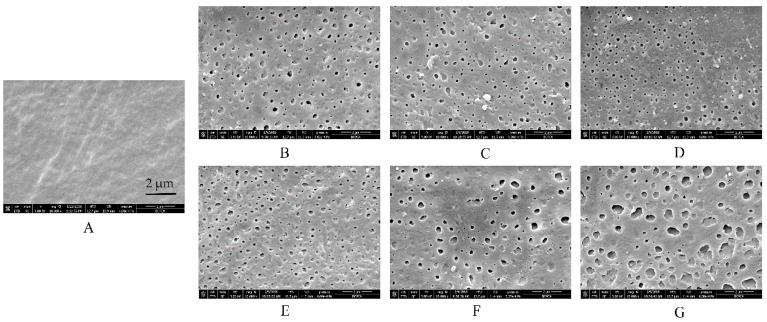
Scanning electron microscope (SEM) images of films containing 20% *w*/*w* HPC (**A**) before and after being submerged in dissolution media with different release buffers for 24 h: (**B**) deionized water, (**C**) HCl solution at pH 1.2, (**D**) phosphate buffer at pH 6.8, (**E**) 0.005 M acetate buffer at pH 4.5, (**F**) 0.05 M acetate buffer at pH 4.5, and (**G**) 0.5 M acetate buffer at pH 4.5.

**Figure 5 pharmaceutics-11-00080-f005:**
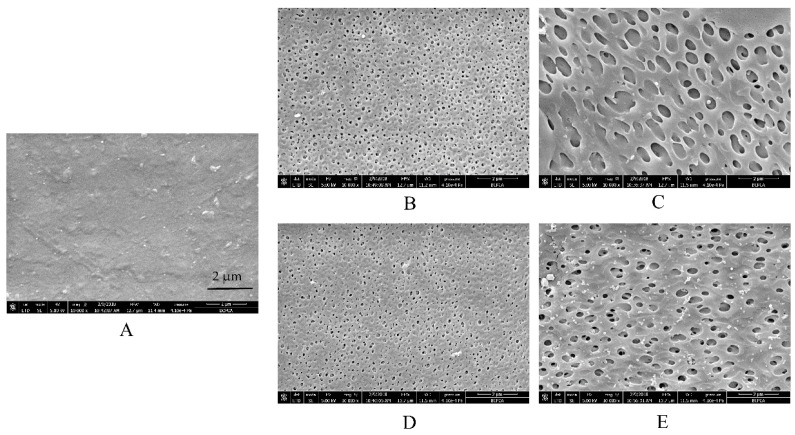
SEM images of films containing 24% *w*/*w* HPC (**A**) before and after being submerged in dissolution media with different release buffers for 24 h: (**B**) deionized water, (**C**) HCl solution at pH 1.2, (**D**) phosphate buffer at pH 6.8, and (**E**) 0.05 M acetate buffer at pH 4.5.

**Figure 6 pharmaceutics-11-00080-f006:**
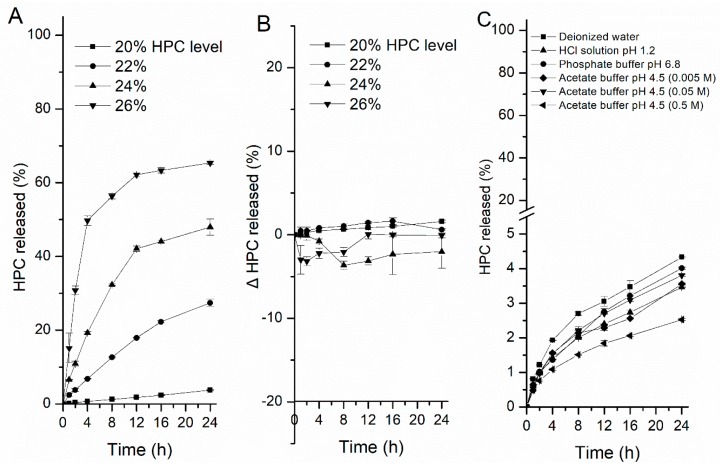
HPC leaching profiles from coated pellets in different media at 37 °C. (**A**) Leaching of HPC from pellets containing 20, 22, 24, and 26% HPC in deionized water. (**B**) The difference between HPC leaching in water and in HCl solution at pH 1.2 with EC: HPC (80:20, 78:22, 76:24, and 74:26) film-coated pellets and (**C**) the release of pellets containing 20% HPC in different media. Data are presented as mean ± SD, *n* = 3.

**Figure 7 pharmaceutics-11-00080-f007:**
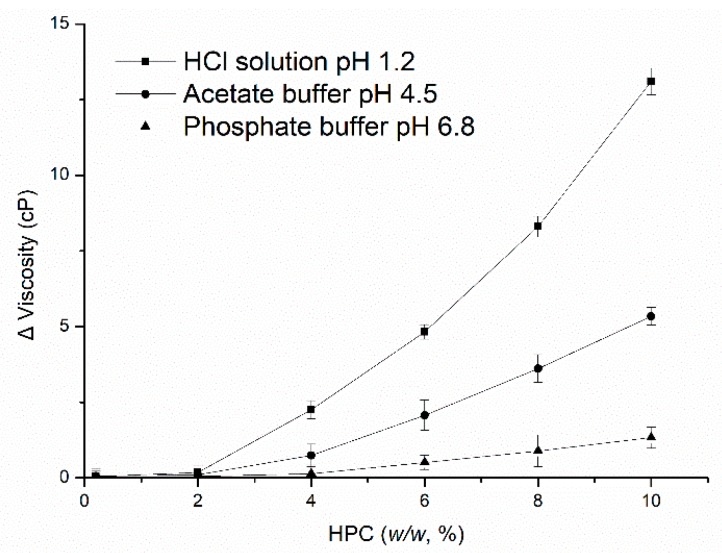
The differences in viscosity of different concentrations of HPC solutions in media at 37 °C, with water as a contrast. Data are presented as mean ± SD, *n* = 3.

**Figure 8 pharmaceutics-11-00080-f008:**
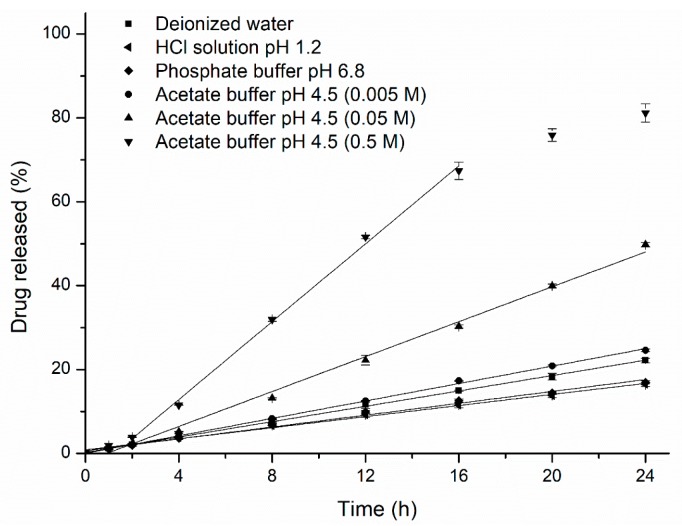
Effects of the pH and ionic strength of the release medium (deionized water; HCl solution at pH 1.2; phosphate buffer at pH 6.8; and 0.005 M, 0.05 M, and 0.5 M acetate buffer at pH 4.5) on the release of metoprolol succinate from HPC-free EC-coated pellets. Straight lines represent the region from which zero-order release rates were estimated (*R*^2^ > 0.99). Error bars represent standard deviation.

**Figure 9 pharmaceutics-11-00080-f009:**
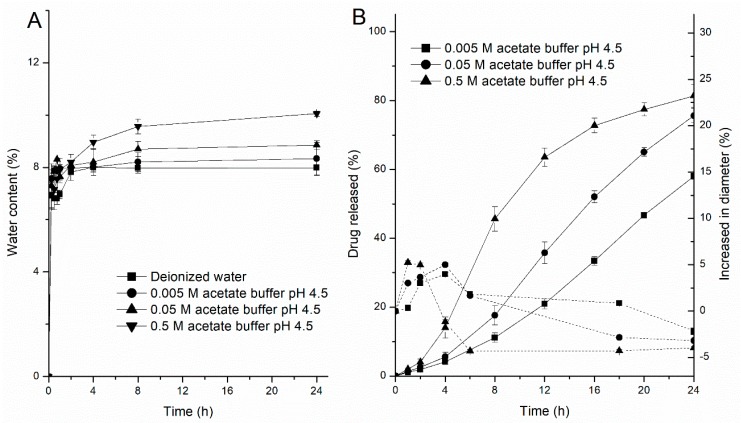
EC/HPC (80:20) films upon exposure to different concentrations of acetate buffer at 37 °C. (**A**) Water uptake kinetics. (**B**) Pellet swelling—on the left *y*-axis, the percentage of drug release is plotted, and on the right *y*-axis, the increase in pellet diameter is shown (in %). Data are presented as mean ± SD.

**Figure 10 pharmaceutics-11-00080-f010:**
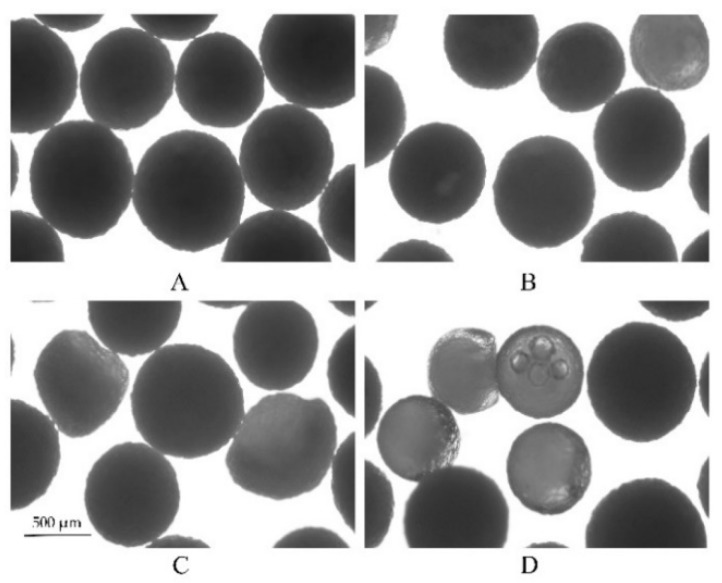
Microscopic pictures of the pellets swelling upon exposure to different concentrations of acetate buffer after 6 h at 37 °C. On the left, the pellets are shown before exposure to the release medium, and on the right, after exposure to: (**A**) 0.005 M acetate buffer at pH 4.5, (**B**) 0.05 M acetate buffer at pH 4.5, and (**C**) 0.5 M acetate buffer at pH 4.5.

**Figure 11 pharmaceutics-11-00080-f011:**
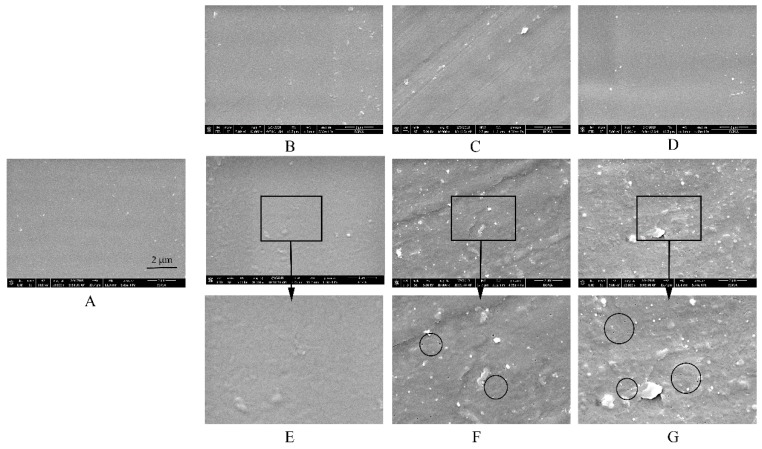
SEM images of pure-EC films (**A**) before and after being submerged in dissolution media with different release buffers for 24h: (**B**) deionized water, (**C**) HCl solution at pH 1.2, (**D**) phosphate buffer at pH 6.8, (**E**) 0.005 M acetate buffer at pH 4.5, (**F**) 0.05 M acetate buffer at pH 4.5, and (**G**) 0.5 M acetate buffer at pH 4.5. The circles area showed the small pores formed in the pure-EC films.

**Figure 12 pharmaceutics-11-00080-f012:**
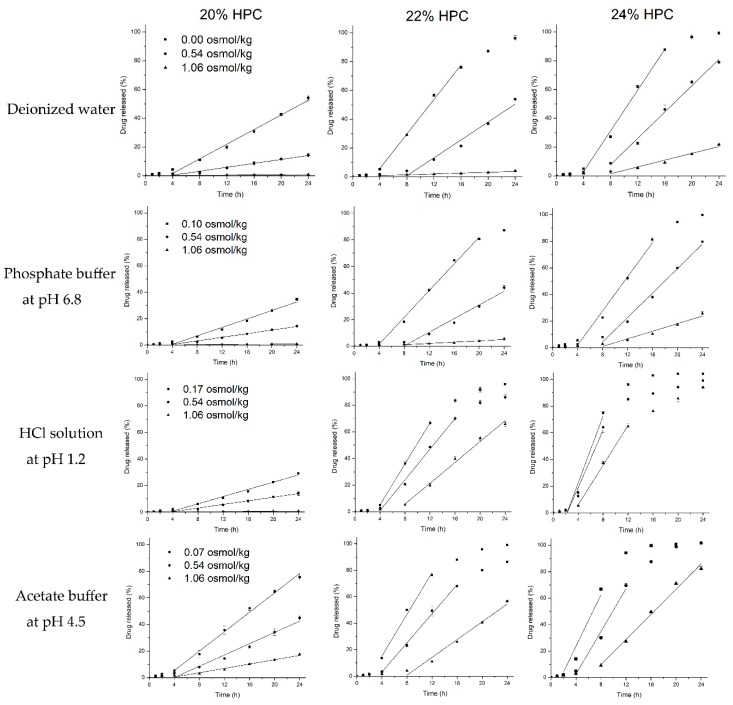
Drug release from EC/HPC (80:20, 78:22, and 76:24 *w*/*w*)-coated pellets in release medium with increasing NaCl concentrations (mOsmol/kg) at 37 °C. Media: deionized water, phosphate buffer solution at pH 6.8, HCl solution at pH 1.2, and acetate buffer solution at pH 4.5. Straight lines represent the region from which the zero-order release rates were estimated (*R*^2^ > 0.98). Error bars represent standard deviation.

**Figure 13 pharmaceutics-11-00080-f013:**
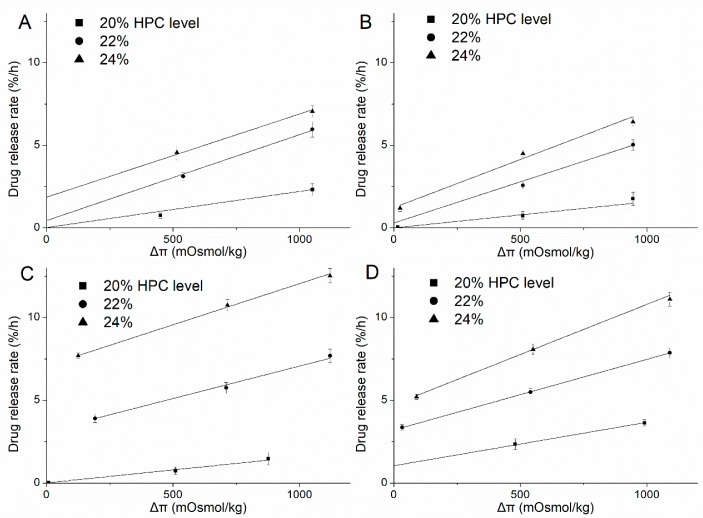
Zero-order release rate at steady state versus the osmolality of the release medium with different ratios of EC/HPC-coated pellets. (**A**) Deionized water. (**B**) Phosphate buffer at pH 6.8. (**C**) HCl solution at pH 1.2. (**D**) Acetate buffer at pH 4.5. Straight lines represent the region from which slopes of linear regression analysis were obtained (*R*^2^ > 0.99).

**Table 1 pharmaceutics-11-00080-t001:** Summary of metoprolol succinate solubility and osmolality in different media at 37 °C (*n* = 3)

Solute	Solubility (mg/mL)	Osmolarity (π) of the Solution (mOsmol/kg)
Deionized water	276.05 ± 3.27	0
HCl solution, pH 1.2	303.53 ± 2.84	174 ± 1.64
Phosphate buffer pH 6.8	249.20 ± 8.69	105 ± 1.81
Acetate buffer pH 4.5 (0.05 M)	268.82 ± 9.21	70 ± 2.13
Acetate buffer pH 4.5 (0.005 M)	259.93 ± 2.24	7 ± 1.12
Acetate buffer pH 4.5 (0.5 M)	304.79 ± 9.62	730 ± 0.95
MS (Saturated solution-water)	-	1050 ± 2.34
MS (Saturated solution-HCl)	-	1294 ± 1.85
MS (Saturated solution-phosphate buffer)	-	1160 ± 2.01
MS (Saturated solution-acetate buffer)	-	1152 ± 2.16

**Table 2 pharmaceutics-11-00080-t002:** Summary of R_T_ (pure EC) and R_T_ (80% EC) determined from linear regression data in Figure 3 and Figure 8.

Medium	R_T (pure EC)_-%-h	R_T (80% EC)_-%-h	R_T (pure EC)_/R_T (80% EC)_
Deionized water	0.758	2.318	0.32
0.005 M, acetate buffer pH 4.5	1.025	2.987	0.34
0.05 M, acetate buffer pH 4.5	2.276	3.632	0.63
0.5 M, acetate buffer pH 4.5	4.680	6.189	0.76

**Table 3 pharmaceutics-11-00080-t003:** Summary of R_T_, Ro and R_D_ determined from the linear regression of R_T_ vs. Δ*π* data in Figure 13

Deionized Water	R_T_-%/h	%/h-(mOsmol/kg)	R_D_-%/h	Ro (R_T_-R_D_)	Ro/R_T_
20%	2.318	0.002	−0.056 *	2.318	1.00
22%	5.966	0.005	0.491	5.475	0.92
24%	7.064	0.006	1.481	5.583	0.79
Phosphate buffer pH 6.8					
20%	1.762	0.002	−0.055 *	1.762	1.00
22%	5.033	0.005	0.241	4.792	0.95
24%	6.418	0.006	1.226	5.192	0.81
HCl solution pH 1.2					
20%	1.465	0.002	−0.011 *	1.465	1.00
22%	7.694	0.004	3.050	4.644	0.60
24%	12.53	0.005	7.135	5.395	0.43
Acetate buffer pH 4.5					
20%	3.632	0.003	1.018	2.614	0.72
22%	7.864	0.004	3.201	4.663	0.59
24%	11.11	0.006	4.954	6.156	0.55

* hydrostatic pressure Δ*P* (see in Equation (4))

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
