# Peer review of "Polymer Distribution and Mechanism Conversion in Multiple Media of Phase-Separated Controlled-Release Film-Coating"

_pharmaceutics, 2019, doi:10.3390/pharmaceutics11020080_

Round 1

Reviewer 1 Report

Dear Author,

Manuscript ID: pharmaceutics-430777 entitled “Polymer distribution and mechanism conversion in multiple media of phase-separated controlled-release film-coating” by Lu Chen, Guobao Yang, Xiaoyang Chu, Chunhong Gao, Yuli Wang, Wei Gong, Zhiping Li, Yang Yang, Meiyan Yang, Chunsheng Gao makes an interesting contribution to current knowledge. It is well organized and typed. The methodology is sound and properly described. The results are respectively presented in the form of tables and figures and the conclusions are justified. In my opinion, the manuscript can be accepted for publication in its current form.

Kind regards,

Author Response

Point: The methodology is sound and properly described. The results are respectively presented in the form of tables and figures and the conclusions are justified. In my opinion, the manuscript can be accepted for publication in its current form.

ResponseThank you very much for your approval to our work.

Reviewer 2 Report

The abstract is poorly written and the english definitely needs to be improved.

The results are missing from the abstract and there is no final conclusion statement for the study.

Line number 73-83, that mentions about the current study needs to be rewritten

Why the author are using UV spectrophotometry as a method of analysis. Do they have reason for it and how are they accounting for interference from other formulation components or polymers. HPLC is more reliable method for assay and would ask to perform estimations using it.

Section 2.3.1-How was drug loading determined?

Section 2.41. and 2.7-mention the predetermined time points.

SEM method is not there and needs to be included in the method section

Most of the figures need to be reworked or replotted. most of the figures are not well drawn, not self explanatory or the data points are cut. Example figure 2 is too tiny and the error bars are not visible. figure 3 the 24 hr time points are cut, figure 3A text in the figure is going out of it and is not consistent.. Figure 4 and Figure 5 SEMs are not good and the scale bars are not visible. Figure 6, 7, 8 data point is cut for 24 hour time point.

figure 12 is hardly visible to understand or interpret any results. Either rework on the figure and improve it or make it into two parts.

Why did the authors choose metoprolol succinate drug and the polymer? need to mention that reason

Author Response

Dear professor,

Thank you for your comments concerning our manuscript entitled “Polymer distribution and mechanism conversion in multiple media of phase-separated controlled-release film-coating” (pharmaceutics-430777). Those comments are all valuable and very helpful for revising and improving our paper as well as the important guiding significance to our researches. We have studied comments carefully and have made correction which we hope meet with approval. Revised portion are marked in the paper. The point to point responds to your comments are listed as following: 

Point 1: The abstract is poorly written and the English definitely needs to be improved.

Response 1: The manuscript has been edited for the language by the ELSEVIER Language Editing Services before submission. We revised the abstract and language again.

Point 2: The results are missing from the abstract and there is no final conclusion statement for the study.

Response 2: The results and final conclusion statements for the study were supplemented in the revised abstract.

Point 3: Line number 73-83, that mentions about the current study needs to be rewritten

Response 3: The current study has been rewritten accordingly.

Point 4: Why the author are using UV spectrophotometry as a method of analysis. Do they have reason for it and how are they accounting for interference from other formulation components or polymers. HPLC is more reliable method for assay and would ask to perform estimations using it.

Response 4: Both HPLC and UV methods are commonly used for the drug assay. In our study, we selected the UV method for rapid determination of drug release. The method validation of the UV method was also conducted including linear, recovery, precision. The results showed the excipients (MCC cores, ethyl cellulose and hydroxypropyl cellulose) were no interference to the measurement and the method was reliable (data not shown). For further confirmation, we also compared the results of the HPLC and UV methods and the data of the two were no significant difference. Of course, the HPLC method was reliable when the excipients were interference to the measurement.

Point 5: Section 2.3.1-How was drug loading determined?

Response 5: The drug content in drug-loaded cores was determined by UV spectrophotometry at a wavelength of 274 nm (UV-1750, Shimadzu, Japan), which was added in the revised manuscript.

Point 6: Section 2.4.1. and 2.7-mention the predetermined time points.

Response 6: The predetermined time points were set as 1, 2, 4, 8, 12, 16, 20, 24 h in section 2.4.1. and 0.25, 0.5, 0.75, 1, 2, 4, 8, 24 h in section 2.7., respectively, which has been added in the revised manuscript.

Point 7: SEM method is not there and needs to be included in the method section

Response 7: The SEM method was shown in section 2.7.

Point 8: Most of the figures need to be reworked or replotted. most of the figures are not well drawn, not self explanatory or the data points are cut. Example figure 2 is too tiny and the error bars are not visible. figure 3 the 24 h time points are cut, figure 3A text in the figure is going out of it and is not consistent. Figure 4 and Figure 5 SEMs are not good and the scale bars are not visible. Figure 6, 7, 8 data point is cut for 24 hour time point.

Response 8: We checked and revised all the above figures, which data point is cut for 24 h time point. The original image we uploaded is appropriate. We think the problems are brought by the imaging inserting into the document.

Point 9: figure 12 is hardly visible to understand or interpret any results. Either rework on the figure and improve it or make it into two parts.

Response 9: Figure 12 has been revised and the explanation of Figure 12 was shown in section 3.5.

Point 10: Why did the authors choose metoprolol succinate drug and the polymer? need to mention that reason

Response 10: Ethyl cellulose (EC) is a kind of water-insoluble natural cellulose derivate with lower toxicity and allergenicity, which is frequently used as film-forming material for film-controlled sustained-release formulations. However, pure EC films are poorly permeable and need to combined with water-soluble polymers. We selected the water-soluble hydroxypropyl cellulose (HPC) as pore-former due to its good film forming properties and can conveniently be co-dissolved with EC in ethanol, then to increase the permeability of the coating films and thus to control the release of drug. In the previous study, it has been reported that drug release from the phase-separated films coated pellets is complicated and influenced by many factors, such as the drug solubility, the type and level of the polymer. Therefore, to obtain a deeper insight into the drug release kinetics from phase-separated films coated pellets, we selected the highly water-soluble metoprolol succinate (MS) (276 mg/ml at 37 in water) as the model drug, which could give further contribute to design and optimization of phase-separated controlled-release film-coating system. To our knowledge, there is little reporting on the MS-release kinetics in multiple media for deeper and comprehensive understand the release mechanisms and the microstructure of the EC/HPC film.

Round 2

Reviewer 2 Report

All comments addressed